# Fast Detection of Heavy Metal Content in *Fritillaria thunbergii* by Laser-Induced Breakdown Spectroscopy with PSO-BP and SSA-BP Analysis

**DOI:** 10.3390/molecules28083360

**Published:** 2023-04-11

**Authors:** Xinmeng Luo, Rongqin Chen, Muhammad Hilal Kabir, Fei Liu, Zhengyu Tao, Lijuan Liu, Wenwen Kong

**Affiliations:** 1College of Mathematics and Computer Science, Zhejiang A&F University, 666 Wusu Street, Hangzhou 311300, China; 2College of Biosystems Engineering and Food Science, Zhejiang University, 866 Yuhangtang Road, Hangzhou 310058, China

**Keywords:** heavy metal, herbal medicine, laser-induced breakdown spectroscopy (LIBS), PSO-BP, SSA-BP, *Fritillaria thunbergii*

## Abstract

Fast detection of heavy metals is important to ensure the quality and safety of herbal medicines. In this study, laser-induced breakdown spectroscopy (LIBS) was applied to detect the heavy metal content (Cd, Cu, and Pb) in *Fritillaria thunbergii*. Quantitative prediction models were established using a back-propagation neural network (BPNN) optimized using the particle swarm optimization (PSO) algorithm and sparrow search algorithm (SSA), called PSO-BP and SSA-BP, respectively. The results revealed that the BPNN models optimized by PSO and SSA had better accuracy than the BPNN model without optimization. The performance evaluation metrics of the PSO-BP and SSA-BP models were similar. However, the SSA-BP model had two advantages: it was faster and had higher prediction accuracy at low concentrations. For the three heavy metals Cd, Cu and Pb, the prediction correlation coefficient (R_p_^2^) values for the SSA-BP model were 0.972, 0.991 and 0.956; the prediction root mean square error (RMSEP) values were 5.553, 7.810 and 12.906 mg/kg; and the prediction relative percent deviation (RPD) values were 6.04, 10.34 and 4.94, respectively. Therefore, LIBS could be considered a constructive tool for the quantification of Cd, Cu and Pb contents in *Fritillaria thunbergii*.

## 1. Introduction

*Fritillaria thunbergii* (Zhebeimu) is a perennial herb in the lily family [1] that is widely recognized for its medicinal properties; its main production areas are the Zhejiang, Jiangsu and Anhui provinces in China [2]. Chinese herbal medicines have been used in China for thousands of years and have spread worldwide [3]. Although herbal medicines are often considered natural and harmless, there are safety concerns, such as the potential of heavy metal overdose and toxicity [4]. Owing to the growth environment and characteristics of the herb species itself, herbs commonly contain a certain amount of harmful heavy metals, such as Pb, Cd, Hg, As, Cu and Cr. The dry bulbs of *Fritillaria thunbergii* are widely used as expectorants and antitussives [5], and their safety has become a matter of considerable concern. The unique biological properties of Fritillaria lead to the accumulation of heavy metals, especially Cd, when grown in acidic soils. Rapid determination of Cd, Cu and Pb contents is important for evaluating the safety and quality of *Fritillaria thunbergii*.

Laser-induced breakdown spectroscopy (LIBS) is an atomic emission spectroscopic technique that uses a laser as the excitation source [6]. Compared with traditional analytical techniques, such as inductively coupled plasma mass spectrometry (ICP-MS) [7], atomic absorption spectrometry (AAS) [8], and inductively coupled plasma atomic emission spectrometry (ICP-ASE) [9], LIBS has many advantages, such as green spectrometry, a fast analysis speed, the elimination of the need for complex sample preparation, and multiple elemental measurements. LIBS has been widely used for quantitative and qualitative analyses in plant materials [10,11,12], animal tissues [13,14], mineral resources [15,16] and industrial applications [17]. Previous studies have provided a basis for the quantitative prediction of heavy metal content in Chinese medicine using LIBS.

The use of models with high accuracy and generalization capabilities is crucial for making predictions using LIBS technology. Partial least squares regression (PLSR) [18], support vector regression (SVR) [19] and the back-propagation neural network (BPNN) [20,21,22,23] are commonly used modeling methods. However, these methods have limitations such as the inability of PLSR to effectively fit nonlinear relationships and the low accuracy of SVR prediction with large data.

BPNN has strong generalization and fitting abilities; however, there are still problems in its applications, such as slow convergence, ease of falling into local minima, and overfitting. In response to these problems, Mohamad et al. [24] proposed a particle swarm optimization (PSO) algorithm to optimize the initial weights of BPNN, which can effectively improve the prediction accuracy of the model. In 2020, the sparrow search algorithm (SSA) was proposed [25]; it has better characteristics than other swarm algorithms in the testing of both single-peaked and multi-peaked functions, with fast convergence speed and high convergence accuracy. Therefore, some experts have applied SSA algorithms to perform BPNN optimization. Wang et al. [26] applied the intelligent optimization SSA to optimize a BPNN model and improve its prediction accuracy.

Thus, to improve the prediction accuracy of BPNNs, this study introduces PSO and SSA to optimize the weights and thresholds of BPNN, forming hybrid algorithm artificial neural network models named PSO-BP and SSA-BP, respectively. In this study, spectral data collected by LIBS were preprocessed and combined with a feature selection algorithm to construct a quantitative analysis model for predicting the content of heavy metals (Cd, Cu and Pb) in *Fritillaria thunbergii*. We compared the accuracies of the BPNN, PSO-BP and SSA-BP models to determine the optimal prediction model. The optimal model was then compared with the PLSR model commonly used for heavy metal prediction in Chinese herbal medicines to determine its practicality and superiority. By exploring a more effective and convenient detection method for heavy metal content, this study provides a basis for safety monitoring and quality evaluation of *Fritillaria thunbergii*.

## 2. Results and Discussion

### 2.1. Determination of the Best Preprocessing Algorithm

The spectral profile is shown in Figure 1, and the raw spectral data contain 1024 variables from 210 to 231 nm. Based on the National Institute of Standards and Technology (NIST) Atomic Spectroscopy Database (ASD), the characteristic spectral lines of heavy metals (Cd, Cu and Pb) were identified in this wavelength range as Cd II (214.44 nm, 226.50 nm), Cd I (228.80 nm), Cu II (217.94 nm, 222.89 nm, 224.26 nm), Pb I (217.00 nm) and Pb II (220. 35 nm). It was observed that the elements of the *Fritillaria thunbergia* sample were relatively complex with a heterogeneous distribution of elemental emission spectra. Therefore, LIBS data need to be further analyzed using chemometric methods.

To determine the optimal preprocessing algorithm, different prediction methods were applied to raw LIBS spectral data. PLSR models were developed using different preprocessed spectral data as input variables. The results of the PLSR models based on the raw and preprocessed data are shown in Table 1. For Cd prediction, the optimal performance with an R_cv_^2^ of 0.950 and RMSECV of 7.294 mg/kg was obtained using the spectra preprocessed with MSC. Similarly, for Cu prediction, the optimal performance with an R_cv_^2^ of 0.978 and RMSECV of 10.951 mg/kg was obtained using the MSC preprocessed spectra. For Pb prediction, the optimal performance with an R_cv_^2^ of 0.883 and RMSECV of 20.370 mg/kg was obtained using the spectra preprocessed with SNV. The optimal input variables were employed for further calculations.

### 2.2. Sensitive Variable Selection

The LIBS spectrum collected in this study contained 1024 variables. These variables have problems with covariance and high dimensionality, which can lead to redundant information and a slow computing speed. To reduce these problems, PCA, SPA and CARS were used for feature selection to achieve dimensionality reduction.

The results of the dimensionality reduction using the PCA algorithm are shown in Figure 2. The predictions for Cd and Cu had the same dimensionality reduction because they have the same preprocessing method, and PCA involves unsupervised learning. As shown in Figure 2a, the first 6 principal components could explain 95.34% of the total spectral data. For the Pb prediction, 9 principal components accounted for 95.33% of the cumulative explained variance, as shown in Figure 2b. In other words, these variables can retain more raw data than the others. Therefore, we selected the sensitive variables for further modeling based on the first six principal components for Cd and Cu and on the first nine for Pb.

The feature variables were extracted using SPA and CARS as shown in Figure 3. Based on the CARS method, 134, 118, and 91 variables were selected for the prediction of Cd, Cu, and Pb, which represented 13.09%, 11.52%, and 8.89% of the total wavelengths, respectively. The sensitive bands selected by SPA for the predictions of Cd, Cu and Pb were 4, 5 and 3, which were only 0.39%, 0.49% and 0.29% of the full variables, respectively.

### 2.3. Model Estimations and Comparisons

#### 2.3.1. BPNN Model Prediction Results

To select the best prediction model, BPNN models for predicting the Cd, Cu and Pb concentrations were developed using the characteristic variables selected by the PCA, CARS and SPA algorithms as input variables. The results of each model operation are shown in Table 2.

For the prediction of Cd, SPA and PCA selected significantly fewer variables than CARS. However, the SPA-BP model performed better with R_p_^2^ = 0.968, RMSEP = 5.951 mg/kg and RPD = 5.74. The PCA-BP exhibited overfitting; therefore, it required optimization. For Cu concentration prediction, the BPNN models based on SPA and PCA selected variables that were only 0.49% and 0.59% of the full spectrum, respectively. Their performance with R_p_^2^ = 0.943, 0.931, RMSEP = 19.367, 21.311 mg/kg and RPD = 4.27, 3.84, respectively. However, they yielded better results than the variables selected by CARS that accounted for 13.09% of the full spectrum. For the prediction of Pb, although the CARS-BP model achieved superior performance (R_p_^2^ = 0.950, RMSEP = 13.835 mg/kg and RPD = 4.55), the number of variables filtered out was 91, which was much larger than the 3 and 9 variables selected for the SPA and PCA algorithms, respectively, resulting in a long modeling duration.

For the three heavy metal predictions, the BPNN models based on the SPA and PCA algorithms were more suitable for the next step of the calculation. This is because these two algorithms greatly simplified the model operations and had a short modeling duration and high accuracy. Because the BPNN can easily fall into local optimal solutions, overfitting and other shortcomings, it can lead to unsatisfactory results, such as the low prediction accuracy of PCA-BP for Cd and Pb concentrations and the relatively poor prediction accuracy of SPA-BP for Pb concentrations. Therefore, further optimization of the BPNN model is required to improve the prediction accuracy.

#### 2.3.2. PSO-BP and SSA-BP Models’ Prediction Results

The PSO-BP and SSA-BP models were established based on variables selected by the PCA and SPA methods, respectively, to further explore their ability to predict heavy metal concentrations in *Fritillaria thunbergii*. The performance of each model is listed in Table 3. It can be seen from the results that the PSO-BP and SSA-BP models had better prediction performance than the raw BPNN model (Table 2). Therefore, the two optimization algorithms enhanced the performance, and the accuracy of the models based on different characteristic variables and predictions of different heavy metals was improved in both cases. 

The performance evaluation metrics of the PCA-PSO-BP and PCA-SSA-BP models were better than those of SPA-PSO-BP and SPA-SSA-BP (Table 3), indicating that PCA is more suitable for joint modeling with PSO-BP and SSA-BP. In addition, the optimal models for all three heavy metal predictions were based on the variables selected using PCA. For the optimal PCA-PSO-BP models of Cd and Pb, R_p_^2^ = 0.982 and 0.964, RMSEP = 4.461 and 11.791 mg/kg, and RPD = 7.72 and 5.47. For the prediction of Cu, the best result was obtained by the PCA-SSA-BP model, and the optimal model had an R_p_^2^ of 0.991, RMSEP of 7.810 mg/kg and RPD of 10.34. Scatter plots of the models with the best results in the prediction set for the prediction of Cd, Cu and Pb are shown in Figure 4.

#### 2.3.3. Comparison with a Conventional Method (PLSR)

We compared the PSO-BP and SSA-BP methods with the commonly used spectral data modeling method, PLSR. Each method used the same data segmentation, preprocessing, and sensitive variable selection methods for modeling, and the results are shown in Table 4. The PLSR method achieved the best results for Cu estimation, as indicated by the following statistical results: R_p_^2^ = 0.951, RMSEP = 17.996 mg/kg and RPD = 4.50. However, compared with the results in Table 3, the prediction accuracy of the PLSR model for Cu was not as good as that of the PSO-BP and SSA-BP models. For Cd and Pb concentration predictions, the prediction results of PLSR were far inferior to those of the two optimization algorithms with the BPNN.

The prediction results of the three models were compared to better visualize the effectiveness of the SSA-BP and PSO-BP prediction models. Figure 5 shows the relative errors between the predicted and true values. For the prediction of Cd and Cu concentrations, the relative errors of both SSA-BP and PSO-BP were lower than those of PLSR at high concentrations, with a better prediction performance of SSA-BP at low concentrations and relatively larger errors of both PSO-BP and PLSR. Regarding the prediction of Pb concentrations, SSA-BP and PSO-BP provided better predictions than PLSR at both low and high concentrations, but the effect of PSO-BP was not obvious. In summary, the two optimization models, PSO-BP and SSA-BP, were found to be more accurate than the commonly used linear PLSR model for the prediction of heavy metals in *Fritillaria thunbergii*.

#### 2.3.4. Comparison of PSO-BP and SSA-BP

As seen in Table 3, the performances of both the PSO-BP and SSA-BP models are good for the prediction of the 3 heavy metals—the values of R_p_^2^ and RMSEP are very similar, and RPD values are greater than 3, indicating that both models are valid and stable. However, the modeling duration of the models differed (the average value of 10 operation times was used as the final reference basis)—the modeling duration of the PSO-BP model was approximately 3 times longer than that of the SSA-BP model; thus, SSA-BP significantly outperformed PSO-BP in terms of computing speed.

Figure 6 and Figure 7 show the relative errors between the first two sets of data predicted by the BPNN, PSO-BP and SSA-BP models and the true values. For the prediction of low concentrations, the relative errors of the SSA-BP model are all lower than those of the BPNN model, whereas the PSO-BP model is not stable for the prediction of low concentrations and is sometimes less accurate than the BPNN model. The relative errors of the SSA-BP model are lower in most cases compared with the PSO-BP model; therefore, the prediction results of the SSA-BP model are more informative in practical use.

In summary, the SSA-BP model was used to predict the concentrations of three heavy metals (Cd, Cu and Pb) in *Fritillaria thunbergii*, and a stable, accurate and practical reference result was obtained in a short period of time.

## 3. Materials and Methods

### 3.1. Fritillaria thunbergii Material and Sampling

Eight different brands of *Fritillaria thunbergii* commonly found on the market were purchased and used in this experiment (Table 5). Samples with different Cd, Cu and Pb contents were prepared in the laboratory by treatment with cadmium nitrate, copper nitrate and lead nitrate solutions. 

Cadmium nitrate, copper nitrate and lead nitrate powders were used to prepare a solution of 0.01 mol/L each. The 8 types of *Fritillaria thunbergii* were crushed using a high-speed crusher and each type of *Fritillaria thunbergii* was equally divided into 8 groups and coded. A concentration gradient (for Cd: 0, 5, 10, 20, 25, 50, 80 and 100 mg/kg; for Cu: 0, 20, 40, 60, 80, 100, 200 and 300 mg/kg; and for Pb: 0, 5, 20, 40, 60, 80, 100 and 200 mg/kg) was predetermined between each group. The purpose was to produce samples whose true values could also be re-examined by 1-8 ICP–MS. Each group contained 5 g of *Fritillaria thunbergii* powder; 1 group was the control group, and the rest were the treatment groups. Different amounts of the solution were added to each group (for Cd cadmium nitrate solution: 22.24, 44.48, 88.97, 111.21, 222.42, 355.87 and 444.84 μL; for copper nitrate solution: 156.25, 312.5, 468.75, 625, 781.25, 1562.5 and 2343.75 μL; and for lead nitrate solution: 48.26, 96.53, 144.79,193.05, 241.31 and 482.63 μL). All samples were dried in an oven at 60 °C and crushed separately using a pulverizer (AQ-180E, Nail, Ningbo, China) to obtain a homogeneous mixture of each group of samples and to ensure consistency in the content of each heavy metal. The ground samples were formed into 1.5 cm diameter pellets using a tablet compressor. For each group, three parallel samples were prepared, and a total of 192 samples were obtained.

### 3.2. LIBS Experimental

The LIBS setup for this experiment consisted of a pulsed solid-state laser (Vlite 200, Beamtech, Beijing, China), a spectrograph (SR500i, Andor, Belfast, UK), an ICCD camera (DH334-18F-03, Andor, Belfast, UK), a movable sample stage (TSA50-C, Zolix, Beijing, China), an X-Y-Z motorized stage (TSA50-C, Zolix, Beijing, China), a stage controller (SC300-3A, Zolix, Beijing, China), a computer, an energy attenuator, a reflector, a lens and a fiber optic.

In our experiment, the Nd: YAG laser was used as an excitation source. The laser beam was focused 2 mm below the samples with a 100 mm focal length optical lens, forming an ablation spot. To increase the signal-to-noise ratio, the following parameters were optimized: λ = 532 nm, energy = 40 mJ, delay = 2 µs, and gate width = 10 µs. The repetition rate was 1 Hz. Each spectrum was cumulatively acquired, and three laser pulses were taken. In total, 7 LIBS spectra were acquired in 7 different positions, yielding 21 (3 × 7) spectra. To reduce the fluctuation between the laser points, the first of the 7 points was removed, and the average of 18 (3 × 6) spectra was recorded as the LIBS data of the sample.

### 3.3. Reference Method for Heavy Metal Content Determination

The concentrations of Cd, Cu and Pb in all *Fritillaria thunbergii* samples, as determined by inductively coupled plasma mass spectrometry (ICP–MS) analysis (ELAN DRC-e, PerkinElmer, Waltham, MA, USA), were used as reference values. Ginseng (GBW10020) and French beans (GBW10021) were used to assure the accuracy of the results. The results of the Cd, Cu and Pb content determination are shown in Table 6.

### 3.4. Sample Division and Spectral Preprocessing

The 8 concentration gradients of *Fritillaria thunbergii* samples in this experiment were numbered 1–8, and each sample contained 3 sets of experimental data, for a total of 192 spectral sets of data. We used 2 randomly selected sets of data from each sample as the training set for model calculation, and the remaining set of data was used as the test set for a total of 128 sets of training data and 64 sets of test data.

Before the calibration process, it was necessary to employ preprocessing to reduce irrelevant information and remove high-frequency noise and baseline variations. Four preprocessing methods, including multiplicative scatter correction (MSC), standard normal variate (SNV), wavelet transform (WT) and Savitzky–Golay smoothing (SG), were applied separately and compared. The coefficient of determination for cross-validation (R_cv_^2^) and the root mean squared error cross-validation (RMSECV) were used as evaluation indicators to select the optimum preprocessing strategies.

### 3.5. Sensitive Variable Selection

The raw LIBS spectrum of a *Fritillaria thunbergii* sample contained large amounts of data, including multiple element spectral lines and unnecessary information, such as redundancy, collinearity, and background information. In some cases, suitable methods can select effective wavelengths containing useful information and reduce the collinearity and high-dimensionality problems of full-spectrum data to reduce the input, develop simpler models, and increase speed. In this study, competitive adaptive reweighted sampling (CARS), successive projection algorithm (SPA), and principal component analysis (PCA) were used to select characteristic wavelengths.

### 3.6. Discriminant Analysis Method

#### 3.6.1. BPNN

A BPNN is self-organizing, self-learning, and self-adaptive, and its principles are simple and easy to implement. BPNNs have been used in several applications. A BPNN is a multilayer feed-forward neural network that includes an input layer, implicit layer, and output layer [27]. The main feature of the BPNN algorithm is that the signal is transmitted forward, and the weights and thresholds of the entire network are continuously adjusted by back propagation to reduce errors in the network output [28].

BPNN can achieve arbitrarily complex nonlinear mapping [29] with a certain degree of robustness and fault tolerance [30]. However, the BPNN algorithm has inherent shortcomings, including unstable training results that tend to fall into local optimum solutions, slow convergence, and data overfitting. Therefore, the initial weights and thresholds of the BPNN algorithm were optimized by introducing the PSO and SSA algorithms with the aim of iterative convergence to the optimal global solution.

#### 3.6.2. PSO-BP

The PSO algorithm is used instead of gradient descent to determine the optimal weights and thresholds of the BPNN [31], which can improve the generalization performance. The PSO algorithm that is combined with the BPNN is called PSO-BP.

PSO, which is a swarm intelligence optimization initially presented by Eberhart and Kennedy (1995) [32], is derived from the predatory behavior of birds in nature. This algorithm is the simplest way for each bird to follow the bird closest to the food and search its surrounding area to obtain food. The basic principle is that each problem solution is considered to be a bird in the search space, called a “particle” whose position, velocity, and fitness values represent the particle characteristics. Assuming that in a *d*-dimensional space, there is an initialized population of potentially optimal particles, the particles adjust their velocity and position according to their individual and global extremes and update their fitness values to achieve the goal of finding an optimal solution.

#### 3.6.3. SSA-BP

The SSA is a novel algorithm with strong optimization capability and fast convergence. The SSA is used to optimize the initial weights of a BPNN, establishing the SSA-BP model, which not only makes full use of the mapping capability of the BPNN, but also has the fast global convergence and learning capability of the SSA. The SSA algorithm that is combined with the BPNN algorithm is called SSA-BP.

The SSA is a novel smart swarm optimization algorithm based on the foraging and antipredatory behavior of sparrows [33]. Sparrows are classified as discoverers, joiners, and scouts in the foraging process [34]. The basic process of SSA is to initialize the sparrow population; calculate individual fitness values and determine the best and worst suited individuals; update the positions of discoverers, joiners, and scouts in turn; and update through successive iterations until the termination conditions are met.

### 3.7. Model Evaluation and Software

The performance of the models was evaluated using seven parameters: the correlation coefficient for calibration (R_c_^2^), cross-validation (R_cv_^2^) and prediction (R_p_^2^), the root mean square error of calibration (RMSEC), the root mean square error of cross-validation (RMSECV) and the root mean square error of prediction (RMSEP), and relative percent deviation (RPD). A good model should have high R_c_^2^, R_cv_^2^ and R_p_^2^ values and low RMSEC, RMSECV and RMSEP values. An RPD greater than 1.5 is considered a good prediction; an RPD between 2.0 and 2.5 indicates that it is a satisfactory prediction model; and an RPD greater than 3.0 is considered a valid prediction model.

Spectral data preprocessing, extraction, and calculation of the SPA, CARS, PCA and PLSR algorithms were performed using Python3.7.3. The BPNN, PSO-BP, and SSA-BP were constructed using MATLAB R2019b (Math Works, Natick, MA, USA).

## 4. Conclusions

This study demonstrated that LIBS can be used to predict the Cd, Cu and Pb contents in *Fritillaria thunbergii*. For the estimation of the content of heavy metals, optimal spectral estimation models were established using BPNN, PSO-BP, SSA-BP and PLSR models. The models established by the PSO-BP and SSA-BP models were better and had higher precision than the other models. The SSA-BP model is superior to the PSO-BP model because it runs faster, has higher prediction accuracy at low concentrations, and is more practical to apply. For the SSA-BP model, Cu had the highest modeling accuracy and perfect prediction, with statistical analysis values of R_p_^2^ = 0.991, RMSEP = 7.810 mg/kg and RPD = 10.34. Cd and Pb were modeled better with R_p_^2^ values of 0.972 and 0.956, respectively; RMSEP values of 5.553 and 12.906 mg/kg, respectively; and RPD values of 6.04 and 4.94, respectively. In summary, LIBS technology is an effective method for the rapid detection of Cd, Cu and Pb in *Fritillaria thunbergii*. This study provides a basis for the application of LIBS to the quantitative determination of heavy metal content in Chinese medicine.

## Figures and Tables

**Figure 1 molecules-28-03360-f001:**
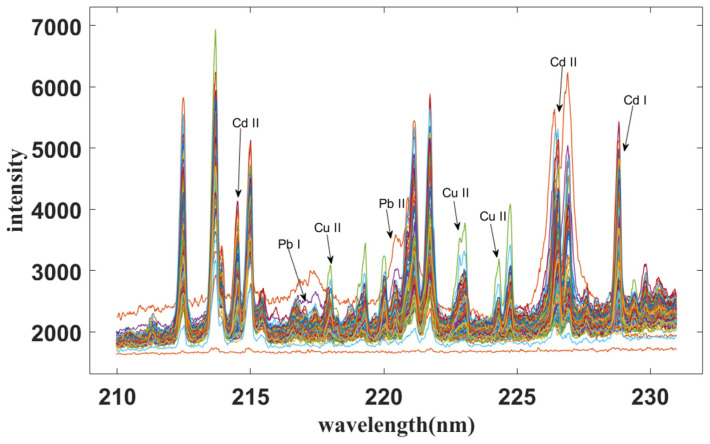
Raw spectra of *Fritillaria thunbergii* measured by the laser-induced breakdown spectroscopy (LIBS) system working in the wavenumber range of 210–231 nm.

**Figure 2 molecules-28-03360-f002:**
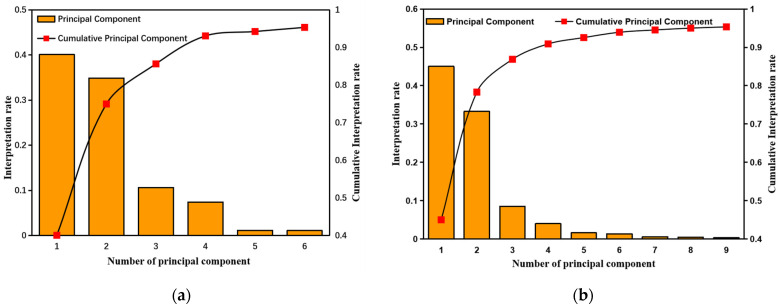
Contribution of each principal component and cumulative contribution of principal components for prediction of (**a**) Cd and Cu; (**b**) Pb.

**Figure 3 molecules-28-03360-f003:**
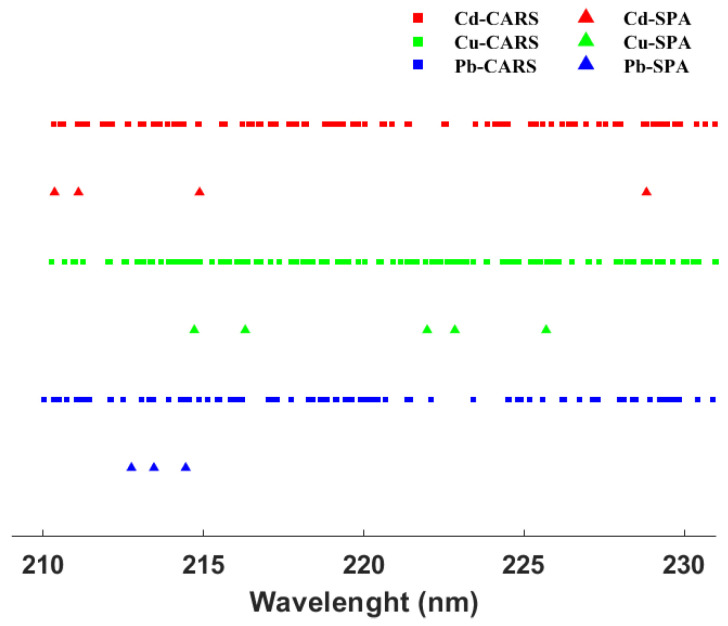
Sensitive variables based on CARS and SPA methods for all heavy metals.

**Figure 4 molecules-28-03360-f004:**
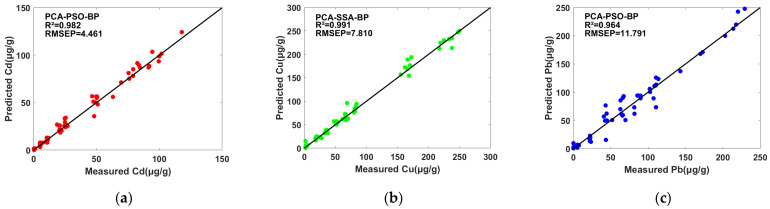
Optimal models on the prediction set. (**a**) Results of the PCA-PSO-BP model for Cd prediction. (**b**) Results of the PCA-SSA-BP model for Cu prediction. (**c**) Results of the PCA-PSO-BP model for Pb prediction.

**Figure 5 molecules-28-03360-f005:**
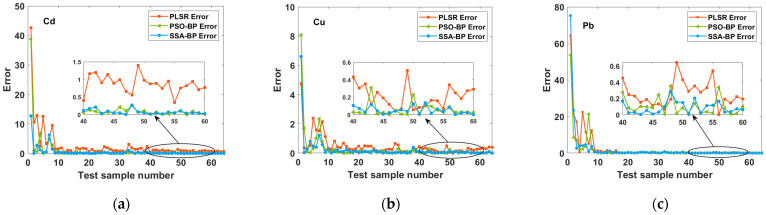
Relative errors of the predicted and true values of heavy metals from PSO-BP, SSA-BP, and PLSR models. (**a**) Relative errors of these three models for the prediction of Cd. (**b**) Relative errors of these three models for the prediction of Cu. (**c**) Relative errors of these three models for the prediction of Pb.

**Figure 6 molecules-28-03360-f006:**
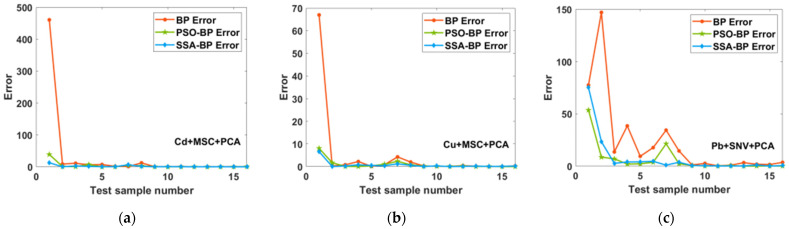
Percentage error of PCA-BP, PSO-BP, and SSA-BP for the first two sets of low concentration data for the prediction of (**a**) Cd; (**b**) Cu; and (**c**) Pb.

**Figure 7 molecules-28-03360-f007:**
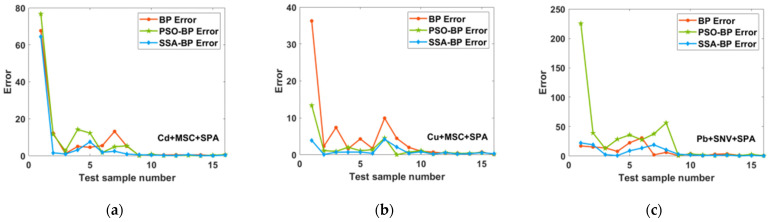
Percentage error of SPA-BP, PSO-BP, and SSA-BP for the first two sets of low concentration data for prediction of (**a**) Cd; (**b**) Cu; and (**c**) Pb.

**Table 1 molecules-28-03360-t001:** Results of partial least squares regression (PLSR) models with different data preprocessing methods.

Preprocessing	Cd		Cu		Pb	
	R_cv_^2^	RMSECV (mg/kg)	R_cv_^2^	RMSECV (mg/kg)	R_cv_^2^	RMSECV (mg/kg)
None	0.932	8.478	0.943	18.021	0.818	25.134
SNV	0.945	7.641	0.974	12.166	0.883	20.370
MSC	0.950	7.294	0.978	10.951	0.872	20.698
WT	0.937	8.205	0.942	17.970	0.823	24.763
SG	0.934	8.393	0.938	18.485	0.809	25.615

**Table 2 molecules-28-03360-t002:** Results of the back-propagation neural network (BPNN) models with different sensitive variable selection methods.

Elements	Models	Number of Latent Variables	Calibration		Prediction		
			R_c_^2^	RMSEC (mg/kg)	R_p_^2^	RMSEP (mg/kg)	RPD
Cd	CARS-BP	104	0.975	5.307	0.914	9.743	3.17
	SPA-BP	4	0.968	5.944	0.968	5.951	5.74
	PCA-BP	6	0.978	4.902	0.743	16.899	2.30
Cu	CARS-BP	134	0.983	10.598	0.650	47.917	2.05
	SPA-BP	5	0.960	16.136	0.943	19.367	4.27
	PCA-BP	6	0.981	11.064	0.931	21.311	3.84
Pb	CARS-BP	91	0.982	8.269	0.950	13.835	4.55
	SPA-BP	3	0.881	21.303	0.849	24.007	2.35
	PCA-BP	9	0.951	13.665	0.889	20.589	3.21

**Table 3 molecules-28-03360-t003:** Results of the PSO-BP and SSA-BP models with different sensitive variable selection methods.

Elements	Models	Modeling Duration (s)	Calibration		Prediction		
			R_c_^2^	RMSEC (mg/kg)	R_p_^2^	RMSEP (mg/kg)	RPD
Cd	PCA-PSO-BP	62.33	0.988	3.643	0.982	4.461	7.72
	PCA-SSA-BP	15.09	0.989	3.551	0.972	5.553	6.04
	SPA-PSO-BP	57.83	0.976	5.138	0.971	5.658	5.88
	SPA-SSA-BP	13.52	0.975	5.264	0.970	5.753	5.82
Cu	PCA-PSO-BP	69.30	0.996	5.013	0.988	9.038	9.16
	PCA-SSA-BP	20.73	0.994	6.081	0.991	7.810	10.34
	SPA-PSO-BP	57.15	0.971	13.816	0.970	13.963	5.83
	SPA-SSA-BP	17.43	0.971	13.808	0.966	14.960	5.44
Pb	PCA-PSO-BP	61.49	0.974	10.042	0.964	11.791	5.47
	PCA-SSA-BP	18.06	0.975	9.790	0.956	12.906	4.94
	SPA-PSO-BP	84.03	0.927	16.724	0.884	21.057	2.89
	SPA-SSA-BP	26.68	0.899	19.671	0.882	21.208	2.83

**Table 4 molecules-28-03360-t004:** Results of PLSR models for the prediction of Cd, Cu, and Pb.

Elements	Calibration		Prediction		
	R_c_^2^	RMSEC (mg/kg)	R_p_^2^	RMSEP (mg/kg)	RPD
Cd	0.960	6.641	0.877	11.671	2.86
Cu	0.954	17.453	0.951	17.996	4.50
Pb	0.879	21.514	0.847	24.181	2.56

**Table 5 molecules-28-03360-t005:** Information on different brands of *Fritillaria thunbergii* samples.

Number	Brand	Origin	Manufacturer
1	Bing Ran	Zhejiang, China	Changchun Fangyitang Economic and Trade Co., Ltd.
2	Xian Weng Song Bao	Zhejiang, China	GuangDong TianCheng Traditional Chinese Medicine Co., Ltd.
3	Chuan Cheng	Zhejiang, China	JiangSu ChuanCheng Traditional Chinese Medicine Co., Ltd.
4	Hui Jing Hua Zun	Zhejiang, China	Bozhou Huijingtang Biotechnology Co., Ltd.
5	Yong Gang	Zhejiang, China	Yonggang Pieces Factory Co., Ltd.
6	Shen Yue	Zhejiang, China	Tonghua Sanbao Ginseng Antler Trading Co., Ltd.
7	Kang Mei	Zhejiang, China	Kangmei Pharmaceutical Co., Ltd. (Guangdong)
8	Yi Ling	Zhejiang, China	Shijiazhuang Yiling Herbal Pieces Co., Ltd.

**Table 6 molecules-28-03360-t006:** Inductively coupled plasma mass spectrometry (ICP–MS) measurement of heavy metal (Cd, Cu, and Pb) content in *Fritillaria thunbergii* samples with eight different treatments (range: the range of the heavy metal content, mean ± SD: the mean and standard deviation of the heavy metal content).

NO.	Cd		Cu		Pb	
	Range (mg/kg)	Mean ± SD (mg/kg)	Range (mg/kg)	Mean ± SD (mg/kg)	Range (mg/kg)	Mean ± SD (mg/kg)
1	0.25–1.2	0.4 ± 0.3	1.9–4.7	3 ± 1	0.16–0.55	0.3 ± 0.1
2	5.0–7.1	5.5 ± 0.6	18–28	21 ± 3	4.1–7.5	5.5 ± 1.0
3	9.9–12	11.0 ± 0.5	33–38	36 ± 2	22–24	22.4 ± 0.5
4	19–27	22 ± 3	48–70	57 ± 8	41–110	50 ± 20
5	21–27	25 ± 2	51–69	65 ± 5	45–70	63 ± 7
6	47–51	49 ± 2	81–84	82 ± 1	63–90	83 ± 8
7	63–120	80 ± 15	160–370	190 ± 70	100–110	110 ± 4
8	84–100	93 ± 6	220–250	230 ± 10	140–230	200 ± 30

## Data Availability

The data, models, and code generated and/or employed during the study are available from the corresponding author upon request.

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
