# Peer review of "Fast Detection of Heavy Metal Content in Fritillaria thunbergii by Laser-Induced Breakdown Spectroscopy with PSO-BP and SSA-BP Analysis"

_molecules, 2023, doi:10.3390/molecules28083360_

Round 1

Reviewer 1 Report

This study provides a basis for the application of Laser-induced breakdown spectroscopy (LIBS) to the quantitative determination of heavy metal content in Chinese medicine. LIBS was applied to detect the heavy metal content (Cd, Cr, and Pb) in Fritillaria thunbergii. Quantitative prediction models were established using the back propagation neural network (BPNN) optimized by a particle swarm algorithm and a sparrow algorithm (PSO-BP and SSA-BP). The manuscript is well-written, however, the following points need to b addressed:

-The method of preparation of samples with different cadmium, copper and lead contents by treatment with cadmium nitrate solution, copper nitrate solution and lead nitrate solution is not sufficiently clear.

-Lines 91-93: Please specify the unit

-Lines 95-96: Please specify the amounts added.

-Line 97: What is the concentration range of Fritillaria thunbergii samples used? You used 5 g powder in all samples. Thus it would be more convenient to specify the range of heavy metals (Cd, Cu and Pb) content in Fritillaria thunbergii samples rather than the concentration range of Fritillaria thunbergii samples. 

-The manuscript needs to be revised by a native English speaker for English language.

Reviewer 2 Report

The basic flaw of this work, prior discussion of the ML algorithms, is the absence of the spectral assignment of atomic lines in LIBS spectra in the range of 210-231 nm regarding Cd, Cu, and Pb analysis in all Fritillaria thunbergii samples. Therefore, the relevance of the acquired spectral lines in this range is not justified prior the following ML analysis of Cd, Cu and Pb content.

Moreover, the manuscript text should be double-read to avoid broken sentence like "Each spectrum was cumulatively acquired 3 laser pulses was taken with a repetition rate of 1 Hz."

Round 2

Reviewer 2 Report

Minor correction in the Abstract is required:   .... The prediction results of the two models PSO-BP and SSA-BP models were similar..... ??????
